# Study of the Self-Locking Characteristics of the Swing Scraper of an Elliptical Rotor Scraper Pump

**Yang Cao** [1,2]**, Tiezhu Zhang** [1,2]**, Hongxin Zhang** [1,2,*]**, Zhen Zhang** [1,2] **, Jian Yang** [1,2] **and Baoquan Liu** [1,2]

1 College of Mechanical and Electrical Engineering, Qingdao University, Qingdao 266071, China; cy082820@163.com (Y.C.); zhangtz@sdut.edu.cn (T.Z.); rexzz9916@163.com (Z.Z.); yangxiaoming8533@163.com (J.Y.); 17865676007@163.com (B.L.)
2 Power Integration and Energy Storage Systems Engineering Technology Center (Qingdao), Qingdao 266071, China
* Correspondence: zhx@qdu.edu.cn; Tel.: +86-135-7386-5229

**Abstract:** This paper proposes an Elliptical Rotor Scraper Pump (ERSP) to address traditional displacement pump defects, such as complex structures, poor self-sealing, low volume utilization, and considerable noise, etc. The ERSP utilizes a swing scraper instead of one rotor in a conventional multirotor pump or reciprocating moving vanes in a traditional vane pump and can achieve high-pressure output through pressure self-sealing. The swing scraper can divide the working chamber into high- and low-pressure rooms. Due to the small swing amplitude of the scraper, the ERSP has low operating noise. The ERSP rotor with an elliptical shape can significantly improve pump volume utilization, thus, forming a kind of fluid pump without a distribution valve, high pressure, and compact structure, and can work efficiently at high speed. This paper establishes a three-dimensional model and mathematical model for the ERSP, then develops the mathematical relationship between the scraper rotation angle and rotor rotation angle and carries out the simulation analysis based on MSC ADAMS. Then, the self-locking characteristics of the ERSP are studied through a force analysis of the swing scraper. Finally, our research group designs and produces a prototype based on existing research and verifies the superiority of the ERSP and the correctness of the non-self-locking condition through experiments. The results in this paper provide a reference for research on the self-locking characteristics of fluid rotor pumps and engineering optimization, which has great significance to the development of fluid power machinery.

**Keywords:** ERSP; pressure self-sealing; simulation analysis; self-locking characteristics; fluid power machinery

## 1. Introduction

### 1.1. Research Motivation

The pump is a mechanical device that converts the prime mover's mechanical energy into the fluid medium's kinetic energy to convey or pressurize the liquid medium and is regarded as the heart of modern industry. It is widely used in agriculture, the chemical industry, mining, metallurgy, electric power, national defense and military industry, urban municipalities, etc., and is most commonly used in the national economy of general machinery [1–3].

### 1.2. Literature Review

There are many principles and types of pumps. According to how the pressure is applied to the fluid, pumps can be divided into volumetric pumps, power pumps, or electromagnetic pumps. A volumetric pump sucks and discharges the liquid by periodically increasing and decreasing the working volume due to the movement of functional parts [4]. A dynamic pump is driven by vanes that rotate the liquid at high speed and

transfer mechanical energy to the conveyed liquid [5]. In an electromagnetic pump, an energized fluid in a magnetic field flows in a specific direction under the action of electromagnetic force. The electromagnetic pump uses the interaction of the magnetic field and electric current in the conductive fluid to subject the fluid to electromagnetic energy and generate a pressure gradient, thus, pushing the fluid movement of a device. In practice, electromagnetic pumps are mainly used for pumping liquid metals, so they are also called liquid metal electromagnetic pumps [6].

Although many pumps are currently in use, each type of pump has inevitable inherent shortcomings. A plunger pump has high output pressure, efficiency, and long service life [7]. However, a plunger pump requires a higher degree of cleanliness in the working medium. The use and maintenance requirements are high to achieve a certain degree of filtration accuracy to ensure regular operation [8]. Gear pumps are insensitive to dirt and impurities in the fluid, are not prone to jamming and have a simple structure and low cost [9]. However, due to the periodic meshing between the teeth, gear pump flow pulsation is inevitable, which will lead to a significant reduction in the stability of the internal flow field. Primarily, the throttle oil phenomenon directly affects the flow pulsation. Flow pulsation leads to pressure ripples. Pressure vibration triggers pump vibration and noise. In addition, the considerable differential pressure between suction inlet and discharge outlet during gear pump operation leads to unbalanced radial forces. This can bend and deform the gear shaft and increase the wear of the moving parts and the load on the bearings, shortening the service life of the gear pump [10,11]. Roots pumps are easy to manufacture at a low cost, and due to their high pumping speed, they have a significant advantage in the case of a large amount of air pumping volumes over a short period [12]. However, the rotor of a roots pump either has an inherent clearance or the clearance increases as the work proceeds, which results in severe internal leakage and low efficiency. A jet pump has a simple structure and low cost and can pump up sludge or other particulate liquids. Nevertheless, its biggest drawback is low efficiency because it works with severe turbulence and friction, so the maximum efficiency is only approximately 30%. In addition, it requires a high-power ground-power supply system [13]. Electromagnetic pumps have no mechanical moving parts. Their simple and compact structure and high-temperature resistance make it possible to transport substances that traditional pumps cannot transport, such as mercury and nonferrous metals. However, superconductivity challenges and expensive costs prevent their wide application [14].

Due to the shortcomings mentioned above, domestic and foreign scholars for pump innovation and research are emerging. M. Bashiri et al. optimized the shape of a centrifugal impeller based on an improved artificial neural network (ANN) and an evolutionary algorithm of particle swarm optimization (PSO), which substantially improved the efficiency and head of a centrifugal pump [15]. Liu D. et al. proposed a new method to effectively reduce flow fluctuations using a variable angular velocity drive [16]. Subsequently, they designed an external noncircular gear drive based on the flow characteristics of the pump, which eliminated the low-frequency large flow ripple generated by the noncircular rotor. Their pump prototype verified that the external noncircular gear drive can effectively reduce the flow pulsation of elliptical gear pumps, providing theoretical support for designing high-performance pumps with large displacement and uniform flow. Wang J. et al. proposed a new type of roots pump with an asymmetric or eccentric involute rotor. The study results showed that the proposed asymmetric rotor has a significant advantage over a conventional rotor in design flexibility [17]. Wu, Y.-R. et al. proposed a mathematical model and method for rotor profile generation of multistage "IVEC"-type roots vacuum pumps [18]. They used two geometric performance indicators, area utilization and meshing gap area, to evaluate different types of rotor profiles. The performance of conventional vacuum pumps was compared with that of IVEC rotor vacuum pumps through experiments. The new rotor shape offered better design flexibility, area utilization, and pump performance than traditional rotor lines to meet the needs of dry multistage roots vacuum pumps. Hsieh et al. proposed a new elliptic profile that was tested using six new rotor

profiles based on elliptic axis ratios. Their results showed that smaller elliptic axis ratio parameters yielded better flow characteristics [19].

The above improvements have played a particular role in promoting the development of pumps. Nevertheless, most of them are secondary innovations based on the original structures, without breaking the barriers of traditional pump disadvantages. W. Xiaoming et al. proposed a new water-hydraulic vane pump that transforms the radial motion of the vane into axial motion, thus, reducing the size of the pump with smooth displacement, excellent motion characteristics, and low noise [20]. Shim, S et al. proposed a new type of volumetric rotary pump, the rotary clap pump. The rotary clap pump is a positive displacement reciprocating pump action converted into a rotary mechanism to reduce the working vibration and the required power [21]. They subsequently performed a kinematic analysis of the operating principle. They verified experimentally that the pump produces relatively low-pressure pulsation, increasing the displacement with less vibration, less power loss, and a significant increase in total efficiency. The rotary clap pump is more suitable for high viscosity and high-flow-rate fluids than other positive displacement pumps [22]. Keisar, D. et al. proposed a new positive displacement, high-pressure, vertical shaft wind pump (HP-VAWP) and experimentally proved that the HP-VAWP has the advantages of zero-carbon emission, high efficiency, and simple manufacturing and maintenance [23]. Wei, X et al. proposed a new reciprocating piston pump rotary sleeve flow distribution system, which solved the disadvantages of traditional piston pumps, such as significant throttling losses, low efficiency, and the tremendous influence of operating frequency [24]. To achieve precise flow control and broaden the application range of flow pumps, Lu, Q. et al. designed a new type of precision flow pump using super magnetostrictive materials as the driving source [25]. The pump adopts an i-shaped pump body, which simplifies the structure and enhances the sealing of the system simultaneously, making the pump more adaptable to miniaturized volume requirement occasions. Ivanovi, L. et al. used the Taguchi method to analyze the influence of the coefficient of the trochoid radius, the number of revolutions and the working pressure on the change in the flow rate and the volumetric efficiency of the trochoidal rotary pump with internal gearing. The conformation of the experiment shows that the Taguchi method can be successfully used for selection of the optimal combination of parameters so that the maximum volumetric efficiency of the trochoid pump can be achieved. This research has guiding significance to increase the volumetric efficiency of rotor pumps [26].

The approach of the above study is of general interest, but the problems related to the lack of compactness pump structures and the low displacement per unit volume remain to be solved. In addition, many applications of rotor pumps tend to have fast response, small magnitude, low noise, and high flow direction, but all lack reliable self-sealing, complex structure, processing, and assembly difficulties. Based on the above issues, our research group proposes the structural principle of the swing scraper pump, whose simplest structural form is the elliptical rotor scraper pump, with a self-sealing pressure function between the swing scraper and elliptical rotor. The ERSP is a new displacement pump with no need for a flow distribution valve, compact structure, and high efficiency. Pulsation-free flow output can be achieved using appropriate rotor profiles (e.g., a three-blade rotor), multiple scrapers, and inlets and outlets.

The study of self-locking characteristics is a crucial part of power transmission design. From the definition of self-locking, it is known that the movement angle of the swing scraper (the residual angle of the pressure angle) being less than the friction angle will make the mechanism self-locking. The mechanism cannot move, no matter how large the driving force is [27]. The phenomenon of self-locking has a wide range of applications in daily life. Lin R et al. proposed the concept of a reconfigurable parallel mechanism based on frictional self-locking composite joints, which can convert trusses and mechanisms [28]. The system provides an infinite continuous locking structure, high load, and impact resistance and reduces energy consumption. This self-locking-jointed parallel robot's concept and design approach have many applications in robots with high load or impact resistance.

As a piece of fluid power machinery, the pump needs to maintain stability and fluidity in operation. The self-locking phenomenon is one of the crucial aspects to avoid and prevent in the design. The self-locking of the pump can drastically reduce its efficiency, generate noise, cause severe component wear, and even jam and shorten its service life. This paper aims to study the self-locking characteristics of the ERSP. The proposed new elliptical rotor scraper pump is modelled and analyzed for self-locking characteristics, and experimental verification is given. This paper aims to study the self-locking aspects of the ERSP, modeling and theoretical analysis of the proposed new elliptical rotor scraper pump, and provide experimental verification. The research content of this paper offers solid academic guidance for the development and application of the ERSP, which is of great significance to the field of fluid mechanical transmission.

*1.3. Scientific Contribution of the Paper*

- This paper proposes an elliptical rotor scraper pump (ERSP) and establishes the 3-D model and mathematical model of the ERSP.
- In order to provide a set of basic criteria for the normal operation of ERSP, this paper analyzes the self-locking characteristics of ERSP.
- We calculate the minimum length of the swing scraper according to the self-locking characteristics, and the volume utilization ratio of ERSP is improved.
- Our research group developed a prototype that verifies the rationality of ERSP and the correctness of related research.

*1.4. Organization of the Paper*

The other chapters of this paper are as follows: Section 2 describes the structure and working principle of the ERSP. Section 3 establishes the mathematical model of the ERSP and derives the equations of motion for the scraper. Section 4 carries out a simulation analysis of the ERSP. Section 5 analyses the force analysis and self-locking characteristics of the ERSP scraper. Section 6 shows a physical drawing of the prototype. Section 7 concludes the text and provides an outlook on the work to come.

## 2. Structure and Working Principle

*2.1. 3-D Model of the ERSP*

The 3-D model of the ERSP is shown in Figure 1. The pump operates by converting the input mechanical energy into an output of fluid hydraulic energy. The swinging scraper divides the pump chamber into low-pressure and high-pressure rooms and achieves high-speed and high-efficiency outcomes through pressure self-sealing. The ERSP adopts an elliptical rotor shape, using the swing scraper instead of a reciprocating moving vane, significantly improving the volume utilization rate of the pump. This pump has a simple structure, easy processing of parts, and high economy, which gives it high promotion value [29].

The ERSP mainly consists of an inlet-fluid chamber, outlet-fluid chamber, elliptical rotor, pump body, swing scraper, scraper shaft, compression spring, snap ring, driveshaft, and other parts.

*2.2. Working Principle of the ERSP*

The ERSP works through an external motor driving the drive shaft to drive the cam rotor synchronous rotation, swinging the scraper swing around the pin. Under a compression spring and high-pressure fluid, the back of its tip or low-pressure side is constantly pressed against the elliptical rotor. The higher the fluid pressure is, the greater the force between the swing scraper and the wheel rotor, and the better the sealing effect. At the same time, the volume of the high-pressure chamber decreases, and the high-pressure fluid is output from the outflow pipe, while the magnitude of the low-pressure chamber increases, and the low-pressure fluid enters the low-pressure chamber from the inflow pipe, as shown in Figure 2. The elliptical rotor keeps rotating, the low-pressure fluid keeps

entering the pump chamber, and the high-pressure fluid keeps outputting, transforming mechanical energy into fluid pressure energy. The swing scraper can be designed in various shapes according to the need, and the compression spring can be a spiral spring or torsion spring, etc.

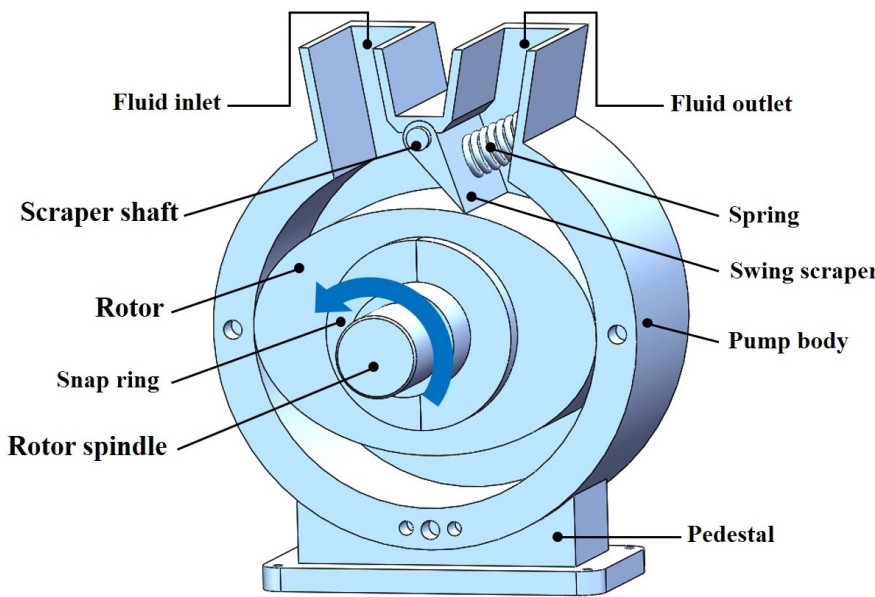

**Figure 1.** The 3-D model of the Elliptical Rotor Scraper Pump (ERSP).

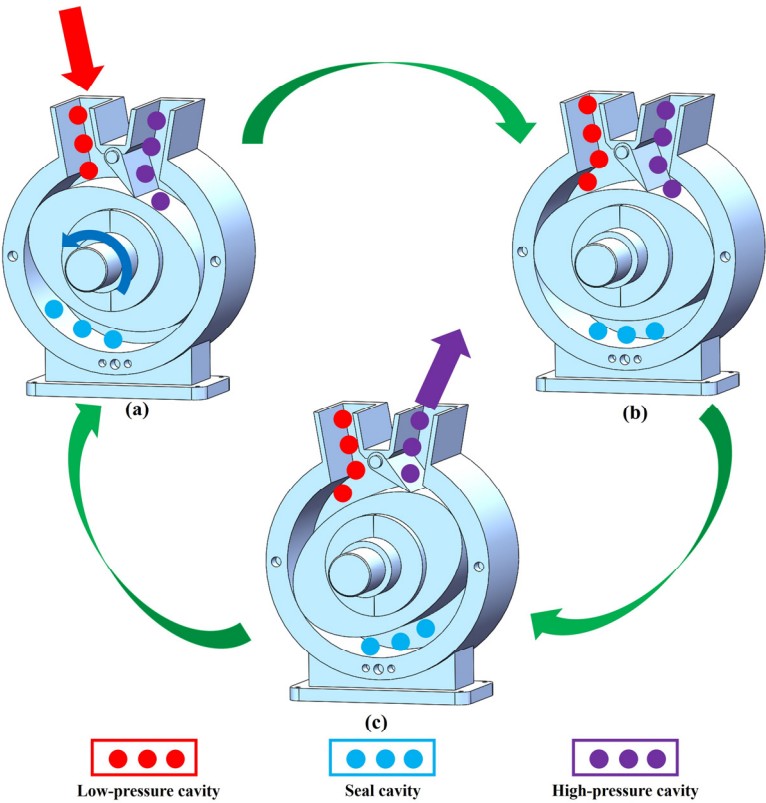

**Figure 2.** Working principle of the ERSP. (**a**) Admission state (**b**) intermediate state (**c**) exhaust state.

Based on the above analysis, the pump proposed in this paper has advantages in terms of flow rate and compression ratio. Gear pumps have tiny gaps when the gears mesh, which can leak fluid and reduce flow. ERSP has good sealing performance, a simple

structure, and does not easily leak fluid, so the flow rate is large. The closed cavity of the vane pump is small and the volumetric efficiency is low, so the compression ratio is low. The overall structure and space of the ERSP are reasonably arranged, the volumetric efficiency is good, and the compression ratio is high.

## 3. Mathematical Model and Kinematic Equation

Self-locking is a phenomenon where, in some cases, the friction force system balances the central power system, and within a specific range, it cannot be made to move, no matter how large the driving force is due to the presence of the friction force. This range is the friction angle, and the condition of self-locking is that the driving force acts within the friction angle. That is, self-locking occurs when the movement angle of the swing scraper is less than the friction angle. When the lower surface of the scraper is tangential to the elliptical rotor, both sides of the scraper will be subject to fluid pressure, and no self-locking phenomenon will occur at this time. Therefore, the mathematical model below discusses the working conditions when the elliptical rotor rotates counterclockwise from the horizontal state to the last contact with the swing scraper tip.

### 3.1. Establishment of Mathematical Model

The swing scraper and elliptical rotor are the critical components in the pump. To analyze the self-locking characteristics of the rotor pump, we must first establish a mathematical model of the pump's operation and clarify the mathematical relationship between the rotor and the rotation angle of the scraper. The following section examines this work in depth.

Figure 3 shows the auxiliary diagram of the mathematical model of the ERSP, where circle A is the trajectory of the swing scraper tip and ellipse O represents the rotor; Q is the intersection of the swing scraper tip and the elliptical rotor at the initial position, Q'', is the intersection at the last contact, and Q' is the intersection at any moment of connection. Figure 3a shows the ERSP's rotor in the horizontal position and the last contact with the tip of the swing scraper. This figure simplifies the swing scraper into a straight line from the center of the upper end of the scraper to the lower tip. The long axis of the elliptical rotor shown is a, the short axis is b, the line AB between the upper end of the scraper and the rotor when horizontal is l, and the length of the scraper is r. Figure 3b shows a partially enlarged image of the angle between the two swing scrapers in Figure 3a. The angle ψ between the scrapers in the two states in the Figure is the maximum angle of swing scraper rotation. Figure 3c shows a schematic diagram of the initial position of the angle between the scraper and the *y*-axis when the elliptical rotor is horizontal, and the angle is φ at this time. Figure 3d shows a diagram of the angle between the swing scraper and the *y*-axis when the elliptical rotor is rotated at any angle. The tip of the swinging scraper is always in contact with the elliptical rotor, at which the angle is φ'.

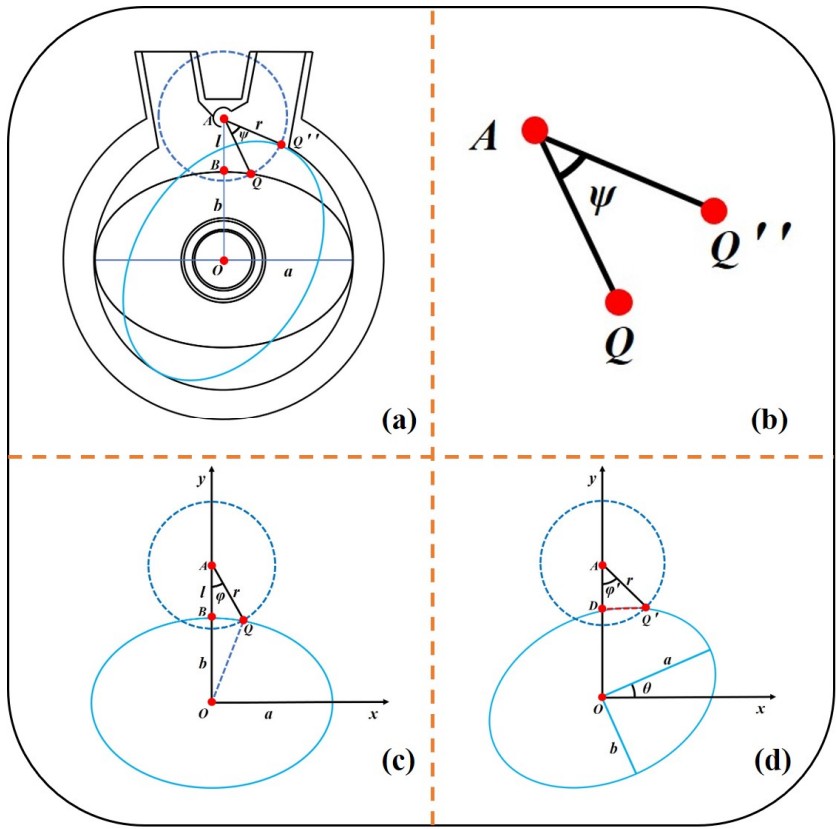

**Figure 3.** Mathematical Model Auxiliary Diagram.

### 3.2. Establishment of Scraper Kinematic Equation

As shown in Figure 3, let the coordinates of point A be (0, b + l), so the equation of a circle with A as the center and r as the radius is as follows:

$$x^2 + [y-(b+l)]^2 = r^2 \tag{1}$$

The equation at the initial state, i.e., when the elliptical rotor is horizontal, is as follows:

$$\frac{x^2}{a^2} + \frac{y^2}{b^2} = 1 \tag{2}$$

When the elliptical rotor is rotated counterclockwise around the center of the circle O by a certain angle, the plane affine transformation equation is:

$$\begin{cases} x' = \cos\theta * x - \frac{a}{b}\sin\theta * y \\ y' = \frac{a}{b}\sin\theta * x + \cos\theta * y \end{cases} \tag{3}$$

The dynamic elliptic equation can be derived as:

$$\frac{x'^2}{a^2} + \frac{x'^2}{b^2} = 1 \tag{4}$$

Combining the equation of the circle A and the ellipse dynamic equation gives:

$$\begin{cases} x^2 + [y-(b+l)]^2 = r^2 \\ \frac{x'^2}{a^2} + \frac{y'^2}{b^2} = 1 \end{cases} \Rightarrow \begin{cases} x_Q = x(\theta) \\ y_Q = y(\theta) \end{cases} \tag{5}$$

Therefore, we can obtain the coordinates of point Q,

$$Q = (x_Q, y_Q) = [x(\theta), y(\theta)] \tag{6}$$

According to the trigonometric function, we can establish the following relationship between $\varphi$ and $\theta$:

$$\tan \varphi' = \frac{x_Q - 0}{b + 1 - y_Q} = \frac{x(\theta)}{b + 1 - y(\theta)} \tag{7}$$

Let $\alpha = \varphi' - \varphi$; then, $\alpha$ is the actual angle of rotation in the scraper. When the tip of the swing scraper is about to leave the elliptical rotor, $\varphi'$ reaches its maximum value, when $\alpha$ is the maximum angle that the swing scraper has turned. The angular size of $\alpha$ affects the operational stability and noise of ERSP, so reasonable control of $\alpha$ should be a focus of ERSP research.

## 4. Simulation Analysis

Based on the theories related to mechanical design and pump design, our research group sized the ERSP for subsequent analysis and prototype development. The detailed parameters are shown in Table 1.

**Table 1.** Basic parameters of the ERSP.

| Parameters | Value |
| --- | --- |
| Long axis of elliptical line a | 741 mm |
| Short axis of elliptical line b | 520 mm |
| Length of swing scraper r | 368 mm |
| Length of the AB l | 297 mm |
| Thickness of elliptical rotor | 228 mm |

In this section, based on the previous theoretical analysis and mathematical model, the kinematic simulation of the motion state of the scraper and rotor is carried out based on MSC ADAMS. During the simulation, we gave the rotor a constant speed of 300 r/min and a pressure of 0.5 atm (approximately 0.05 MPa) on the scraper side of the high-pressure chamber. The simulation starts with the elliptical rotor in the horizontal position, and finally, the first three working cycles of the scraper pump are selected for analysis. Figure 4 is swing scraper rotation angle curve. Figure 5 is rotor and swing scraper rotation angle comparison curve.

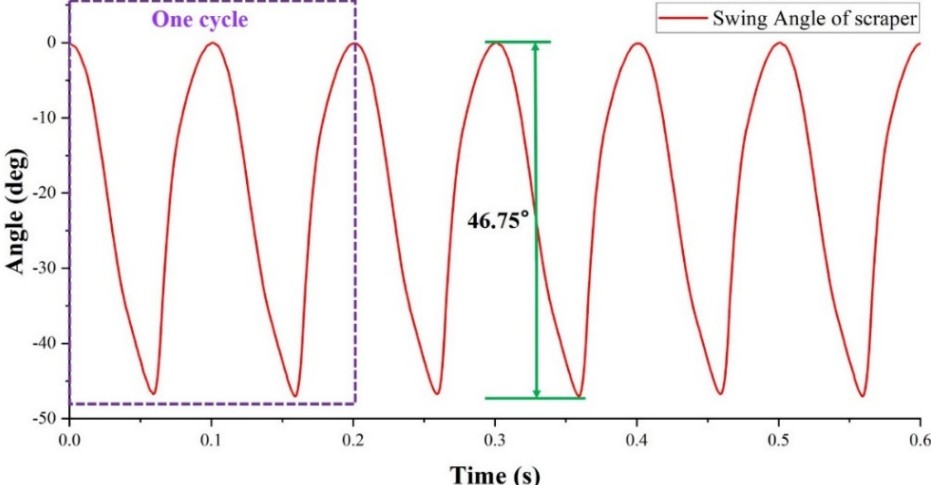

**Figure 4.** Swing scraper rotation angle curve.

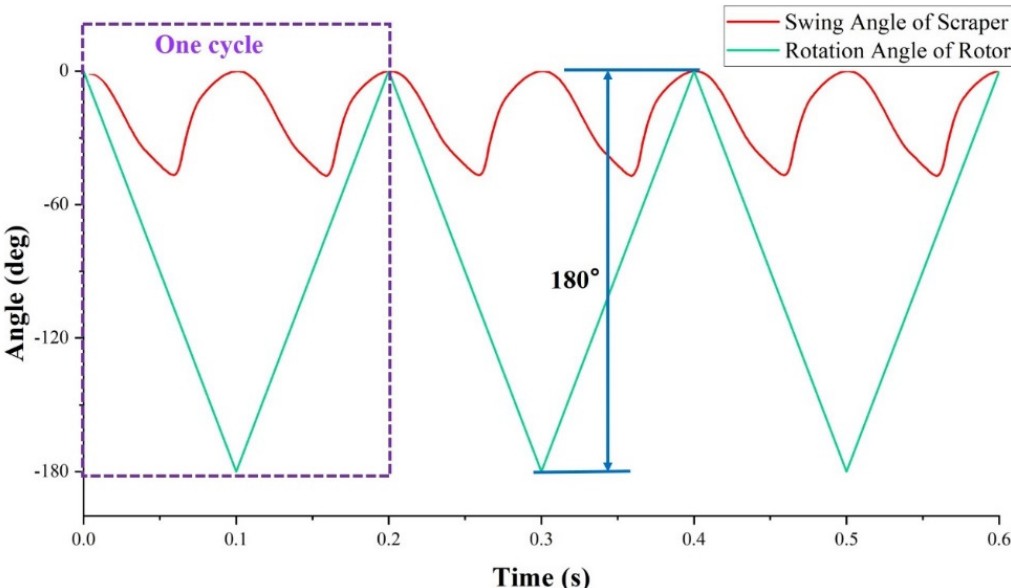

**Figure 5.** Rotor and swing scraper rotation angle comparison curve.

From the simulation analysis results, it can be obtained that the rotor rotates 360° and the scraper swings two cycles accordingly, and the maximum swing angle of the scraper is 46.75°. Meanwhile, the smoothness of the simulated image also proves that the scraper can fully adhere to the elliptical rotor during the operation of the pump and follow well. The maximum rotation angle of the oscillating scraper $\alpha$ = 46.75, and the oscillation amplitude is small, which proves that the ERSP has less noise, lower wear and tear, smooth operation, and longer service life during the working process.

Figure 6 is a graph of the contact force when the elliptical rotor and the swinging scraper move. The maximum contact force is 1266.44 N, and the minimum contact force is 559.96 N. 10,000 points were derived by MSC ADAMS and then fitted to a curve that was in full agreement with the simulation results by MATLAB. Thus, the functional relationship between the contact force and time of the elliptical rotor and the swinging scraper is obtained as:

$$
\begin{aligned}
f(t) = &\ a_0 + a_1 * \cos(t*w) + b_1 * \sin(t*w) + a_2 * \cos(2*t*w) + b_2 * \sin(2*t*w) \\
&+ a_3 * \cos(3*t*w) + b_3 * \sin(3*t*w) + a_4 * \cos(4*t*w) + b_4 * \sin(4*t*w) \\
&+ a_5 * \cos(5*t*w) + b_5 * \sin(5*t*w) + a_6 * \cos(6*t*w) + b_6 * \sin(6*t*w) \\
&+ a_7 * \cos(7*t*w) + b_7 * \sin(7*t*w) + a_8 * \cos(8*t*w) + b_8 * \sin(8*t*w)
\end{aligned}
\tag{8}
$$

Among them: $a_0$ = 848.9, $a_1$ = 1.191, $b_1$ = 0.1147, $a_2$ = 1.149, $b_2$ = 0.2169, $a_3$ = 197.3, $b_3$ = 140, $a_4$ = 1.159, $b_4$ = 0.4999, $a_5$ = 1.078, $b_5$ = 0.5544, $a_6$ = −29.28, $b_6$ = 158.7, $a_7$ = 0.8731, $b_7$ = 0.8868, $a_8$ = 0.7858, $b_8$ = 0.9309, w = 20.95.

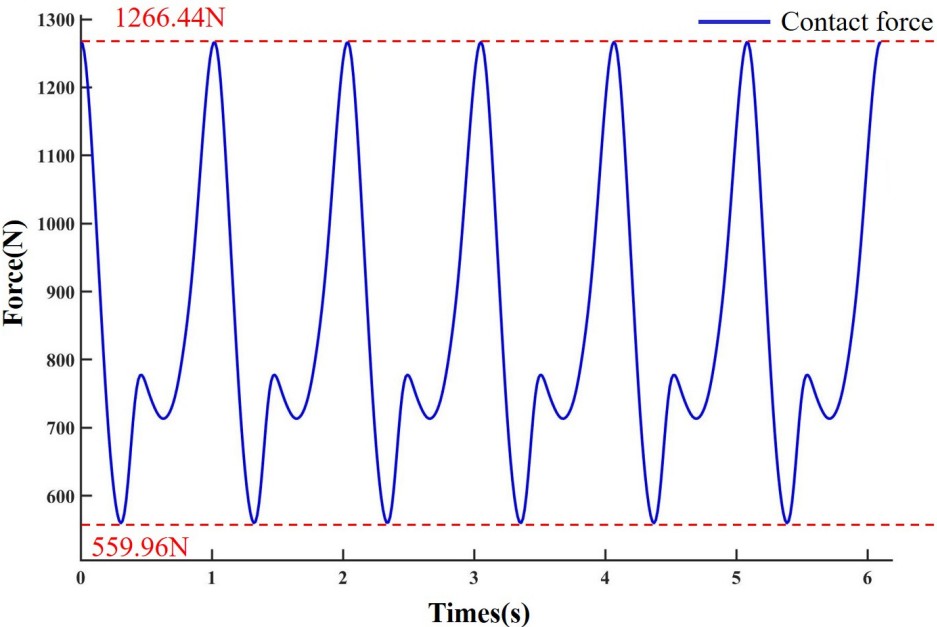

**Figure 6.** Curve of contact force.

### 5. Self-Locking Characteristics Analysis

The force analysis of the swing scraper is shown in Figure 6, including the vertical downwards gravitational force G, the spring compression force $F_K$, the relative fluid pressure $F_L$ perpendicular to the swing scraper, the support force of pin A and the support force $F_N$ of the elliptical rotor for the swing scraper. Among them, the swing scraper gravity and spring compression force are much smaller than the fluid pressure, so they can be neglected. The support force of the pin is decomposed into a component force $F_{A1}$ along the *x*-axis and a component force $F_{A2}$ along the *y*-axis, according to the coordinate system shown in the Figure. $F_N$ is decomposed into a component force $F_{N1}$ along the *x*-axis and a component force $F_{N2}$ along the *y*-axis and a component force $F_R$ along the direction normal to the tangent of the elliptical rotor. The calculation can be performed to determine each component force, so that $\Sigma F_x = 0$ and $\Sigma F_y = 0$. Therefore, the direction of the combined force on the oscillating scraper can be determined as perpendicular to the tangent line of the elliptical rotor facing upwards, with the magnitude $F_R$. In addition, the axial clearance of ERSP is between 0.05–0.1 mm, so the influence of the frictional forces of the sealing surfaces in the axial direction can be completely ignored.

When the elliptical rotor is in the horizontal state, the movement angle of the oscillating scraper is the smallest and will become larger and larger as the elliptical rotor rotates. Therefore, the self-locking characteristic analysis only requires a force analysis of the swing scraper in the horizontal condition of the elliptical rotor.

#### 5.1. Theoretical Formula Derivation

From the theory of self-locking, it is known that the self-locking phenomenon will not occur if the movement angle of the scraper is greater than the friction angle. For the ERSP, the pressure angle is the angle between the direction of the force on the swing scraper and the direction of motion (acute angle), and the movement angle is the residual angle of the pressure angle. As shown in Figure 7, the pressure angle of the swing scraper is $\beta_1$, and the movement angle is $\gamma_1$.

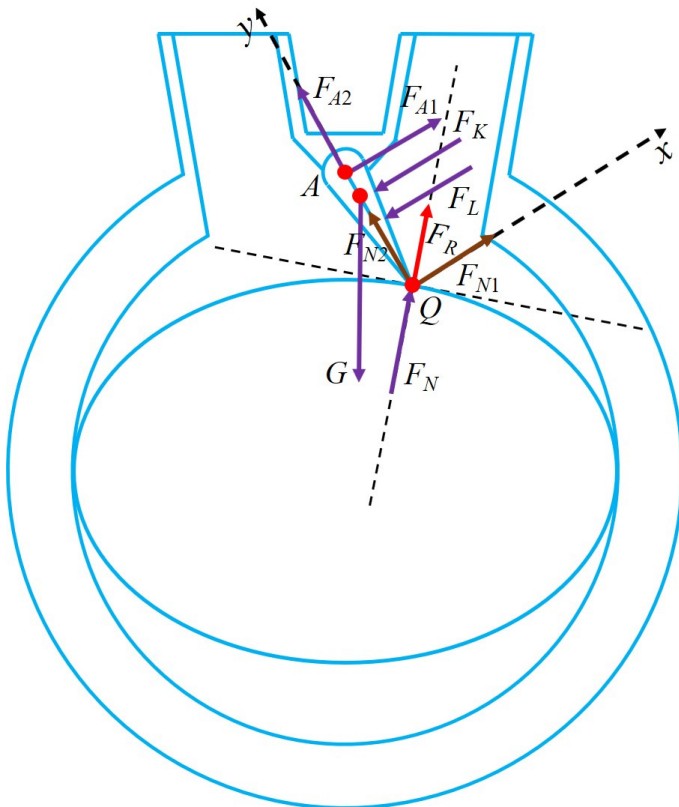

**Figure 7.** Force analysis diagram of the swing scraper.

When the elliptical rotor is in the horizontal state, the direction and magnitude of the combined force on the swing scraper and the direction and magnitude of the velocity are shown in Figure 8a.

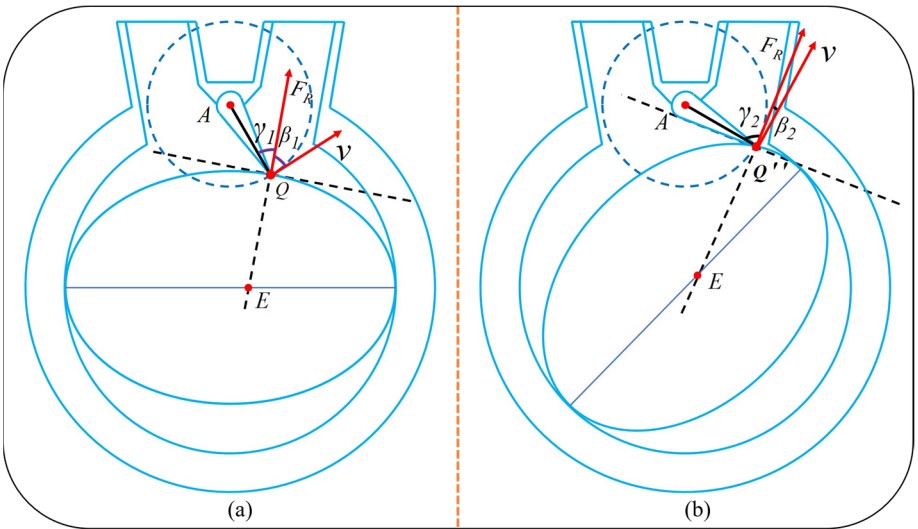

**Figure 8.** Schematic diagram of the swing scraper pressure angle and movement angle in different states of the elliptical rotor. (**a**) initial state; (**b**) critical state.

According to the equation of the elliptic tangent line, we can obtain the equation of the elliptic tangent line through the point Q = $(x_1, y_1)$ as:

$$\frac{xx_1}{a^2} + \frac{yy_1}{b^2} = 1 \tag{9}$$

$QC_1$ is perpendicular to the tangent of the ellipse, so the slope of $QC_1$ is:

$$k_1 = -\frac{a^2 y_1}{1 - b^2 x_1} \tag{10}$$

The ellipse equation at the horizontal is combined with the equation of the circle with A as its center and AC1 as its radius to find the coordinates of the point Q.

$$\begin{cases} \frac{x^2}{a^2} + \frac{y^2}{b^2} = 1 \\ x^2 + [y - (b+1)]^2 = r^2 \end{cases} \tag{11}$$

$$\Rightarrow \frac{b^2 - a^2}{b^2} y^2 - 2(b+1)y + (b+1)^2 - r^2 + a^2 = 0 \quad (y>0) \tag{12}$$

We can find the coordinates of the point $Q = (x_1, y_1)$:

$$\begin{aligned} x_1 &= \sqrt{a^2 - \frac{a^2}{b^2} y_1{}^2} \\ y_1 &= \frac{b^2(b+1) - \sqrt{b^4(b+a)^2 - b^2(b^2 - a^2)\left[(b+1)^2 + a^2 - r^2\right]}}{b^2 - a^2} \end{aligned} \tag{13}$$

The coordinates of point A are known $A = (0, b + 1)$. Therefore, the equation of AQ is as follows:

$$y_{AQ} = \frac{y_1 - b - 1}{x_1} x + b + 1 \tag{14}$$

Therefore, the slope of $QC_1$ is:

$$k_2 = \frac{y_1 - b - 1}{x_1} \tag{15}$$

According to the slope, we can find the pressure angle β1 and movement angle $\gamma_1$ of the oscillating scraper. The specific results are as follows:

$$\beta_1 = \arctan\frac{|k_1 - k_2|}{|1 + k_1 k_2|}, \gamma_1 = \frac{\pi}{2} - \beta_1 = \frac{\pi}{2} - \arctan\left|\frac{k_1 - k_2}{1 + k_1 k_2}\right| \tag{16}$$

At this time, the movement angle of the scraper is minimal, and as long as $\gamma_1 > \varphi$ (where $\varphi$ is the friction angle), the ERSP will not exhibit a self-locking phenomenon.

At the moment when the swing scraper finally comes into contact with the elliptical rotor, the magnitude and direction of the combined force on the swing scraper are shown in Figure 8b. At this time, the angle of motion of the oscillating scraper is at its maximum. When the elliptical rotor moves to tangency with the lower surface of the swing scraper, both sides of the swing scraper are subjected to fluid pressure at this time, so there is no self-locking phenomenon during the whole process of tangency.

From the above analysis, we can easily conclude that the pressure angle of the swing scraper is maximum when the elliptical rotor is horizontal. As the elliptical rotor rotates counterclockwise, the scraper's pressure angle decreases until the pressure angle reaches a minimum at the last moment of contact between the elliptical rotor and the tip of the oscillating scraper. Correspondingly, the movement angle increases. Therefore, as long as the movement angle of the scraper in the horizontal state of the elliptical rotor is greater than the friction angle, the ERSP will not be self-locking in any case. The above work can provide theoretical guidance for the future design and manufacture of different types of ERSP to avoid the phenomenon of self-locking.

### 5.2. Example Analysis

Now, using 45 steels as the rotor and scraper material, the data can be checked to obtain a 45-steels friction coefficient of 0.182, so the friction angle $\varphi = \arctan 0.182 = 10.31°$.

According to Equation (12), we can obtain $x_1$ = 191 mm and $y_1$ = 502 mm. According to Equations (9) and (14), we can obtain $k_1$ = 5.34 and $k_2$ = −1.65. According to Equation (15), we can obtain $\beta_1$ = 41.83° and $\gamma_1$ = 48.17°. Therefore, the minimum movement angle of the swing scraper is 48.17°. The coefficient of sliding friction of 45 steels with lubrication is 0.182, and the corresponding friction angle is 10.31°. The friction angle is less than the minimum movement angle, so this ERSP model will not produce a self-locking phenomenon. In addition, the contact between the oscillating scraper and the elliptical rotor can be designed as rolling friction when the friction coefficient will be smaller.

## 6. Prototype Physical Display

Based on an in-depth collection and analysis of a large amount of theoretical knowledge about the pump, our research team proposed a new concept of an elliptical rotor scraper pump (ERSP). We developed a prototype machine under the ongoing efforts of the whole team and successfully put it into use, as shown in Figure 9. The rotor shaft is connected to an AC motor, so that the rotor rotates at a constant speed to realize the synchronous motion of the elliptical rotor and the scraper. The motor model used is Y132 M-4 with a power of 7.5 kW. The experiment proves that the whole working process of the ERSP does not show a self-locking phenomenon, and the running process is smooth, with little noise and good performance.

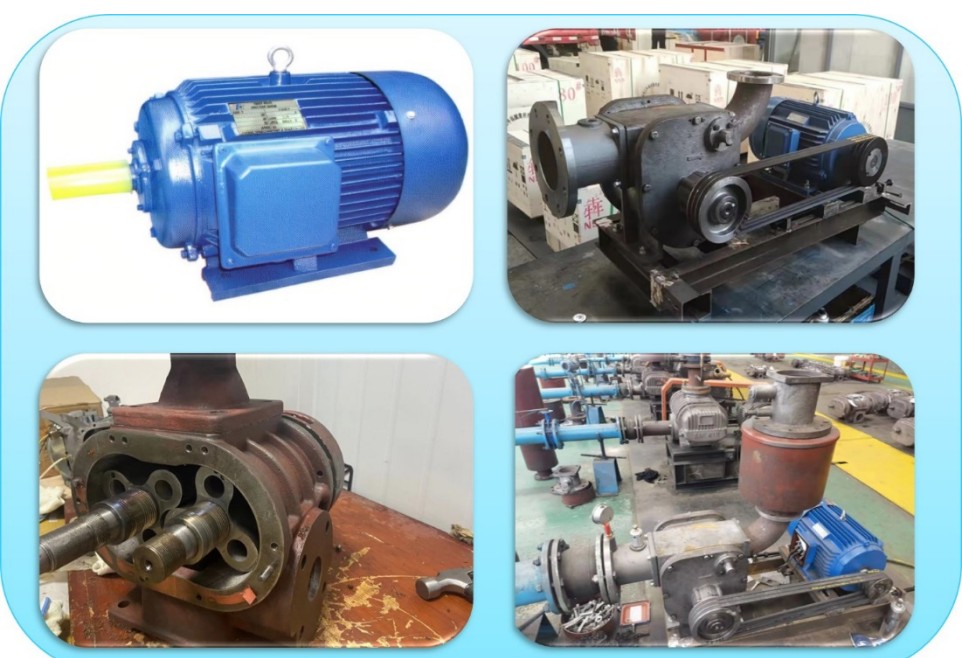

**Figure 9.** Experimental test.

We conducted the corresponding comparative experiment and tested roots pump and ERSP with the same suction diameter. Given the same rotational speed, the experimental results of some important parameters are shown in Table 2. The superiority of the ERSP is further proved by the comparative experiment with the Roots pump.

**Table 2.** Experimental results of roots pump and the ERSP.

|  | Suction Inlet Diameter | Rotary Speed | Flow Rate | Volume Efficiency |
|---|---|---|---|---|
| **Roots Pump** | 100 mm | 1310 r/min | 4.73 m$^3$/min | 72.4% |
| **ERSP** | 100 mm | 1310 r/min | 5.09 m$^3$/min | 83.2% |

## 7. Conclusions and Prospects

This paper proposed a new type of fluid power device, an elliptical rotor scraper pump (ERSP). The ERSP uses a swing scraper instead of one vane of a conventional vane pump. The swing scraper and the elliptical rotor have self-sealing pressure functions so that the ERSP can realize high-pressure output fluid. The ERSP has a simple structure, low requirements for the working environment, and can efficiently achieve the conversion of mechanical energy to fluid energy. Meanwhile, it overcomes the problems of traditional positive displacement pump sealing difficulties, low volume utilization rate, and significant flow pulsation. According to the principle of the ERSP, it is possible to achieve almost no flow pulsation output with an appropriate rotor profile (such as a three-blade rotor), multiple oscillating scrapers, and multiple inlets and outlets.

This paper establishes a three-dimensional model and mathematical model of the ERSP and deduces the mathematical relationship between the elliptical rotor rotation angle and swing scraper rotation angle through theoretical equations. Using MSC ADAMS to simulate the motion of the swing scraper and elliptical rotor, the results show that the following ability of the swing scraper relative to the elliptical rotor is good, as the two parts are always in contact during the movement. The maximum swing angle of the swing scraper is $46.75°$. The swing amplitude is small, which achieves the goal of low noise, fast response and smooth operation of the pump. A force analysis of the swing scraper was carried out, and the self-locking characteristics of the ERSP were studied. The minimum value of the movement angle of the swing scraper is $48.17°$, which is greater than the friction angle, through an example analysis. The research content of this article can perform self-locking judgment on various types of ERSPs to ensure their normal operation and avoid the occurrence of dangerous working conditions. At the same time, the minimum length of the scraper can be determined to reduce the volume of the ERSP, and the volume utilization ratio of the ERSP is improved.

The design of the ERSP is boldly innovative based on the concept of volumetric pump design, improving its structure to maximize volume utilization and, at the same time, providing efficient flow output. In this paper, the structural principle and self-locking characteristics of the ERSP are analyzed more thoroughly, which provides good theoretical guidance for the subsequent design of ERSP's of different sizes. The ERSP has excellent advantages in its structure, self-sealing ability, and efficiency and has excellent prospects for promotion.

**Author Contributions:** Conceptualization, T.Z. and H.Z.; methodology, Y.C. and H.Z.; software, Y.C. and J.Y.; validation, Y.C., Z.Z. and J.Y.; formal analysis, J.Y. and B.L.; investigation, Y.C.; resources, T.Z. and H.Z.; data curation, Y.C. and B.L.; writing—original draft preparation, Y.C.; writing—review and editing, Y.C. and Z.Z.; visualization, B.L.; supervision, Y.C. and J.Y.; project administration, T.Z. and H.Z.; funding acquisition, H.Z. All authors have read and agreed to the published version of the manuscript.

**Funding:** This research was funded by the National Natural Science Foundation of China, grant number 52075278, and the Municipal Livelihood Science and Technology Project of Qingdao, grant number 19-6-1-92-nsh.

**Data Availability Statement:** Not applicable.

**Conflicts of Interest:** The authors declare no conflict of interest.

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
