# Peer review of "Study of the Self-Locking Characteristics of the Swing Scraper of an Elliptical Rotor Scraper Pump"

_machines, doi:10.3390/machines10050370_

Round 1
Reviewer 1 Report
The combination of such elements as Scraper shaft, Spring, Swing seems to be unreliable
Author Response
First of all thanks for your comments.My reply is in the following file.

Reviewer 2 Report
Dear Authors,
The paper describes the elliptical rotor scraper pump, which has been also described in the previous paper Zhang, Z., Zhang, T. Z., Zhang, H. X., Yang, J., Cao, Y., Jiang, Y., & Tian, D. (2022). Modeling, kinematic characteristics analysis 437 and experimental testing of an elliptical rotor scraper pump. Machines, 10(2). The operational parameters like temperature, pressure, flow rate should be described more in details. Following details can be improved:
Line 59: pressure ripples- instead of pressure vibrations.
Line 60 between inlet and outlet- instead import and export measurements.
Figure 1- rotation direction will be helpful.
Line 366,367 The MSC ADAMS results should be presented more in details.

Author Response

(The authors gave the same response as above.)

Reviewer 3 Report
This paper proposes the self-locking characteristics of а swing scraper of an elliptical rotor scraper pump (ERSP). The paper establishes a three-dimensional model and a mathematical model of ERSP, and then a prototype is produced based on existing research. Finally, the superiority of ERSP and the correctness of the nonself-locking condition through experiments were verified. Based on the obtained results, conclusions were drawn.
I believe that this paper can be accepted for publication after a major revision. The following comments should be addressed before publication:
- The scientific contribution of this paper is not clearly highlighted and how the present information would be used to improve functioning elliptical rotor scraper pumps.
- Literature review is not comprehensive. Gerotor pumps is also in line with this field of the research. So, the following references are suggested to cite in the revised paper:
- Doi 10.17559/TV-20150429090420
- It would be useful to make a comparison between the swing scraper of an elliptical rotor scraper pump and similar types of pumps from the viewpoint of self-locking characteristics, flow rate, compression ratio etc.
- The authors claim that the superiority of ERSP and the correctness of the nonself-locking condition have been verified through experiments. However, experimental results have not been presented. Without experimental results, only results from modelling in some mathematical software cannot convince the readers. Therefore, I suggests to add some measurement results to support the modelling results.
- Experimental section could be enriched by an ISO schematic of test rig and the description of the instrumentation.
- In conclusion, give a suggestion on how to use the obtained results.
Author Response

(The authors gave the same response as above.)

Reviewer 4 Report
The abstract of the paper states that a three-dimensional model of the pump (mechanical, hydraulic?) Is presented. There is no 3-D model.
The squeegee equation of motion should be presented in Chapter 3.2. The equations presented in Chapter 3.2 represent a simplified kinematic model of squeegee movement. The equation of motion must contain a correct analysis of the force effects as a function of time. The analysis of force effects presented in Chap. 5 contains major shortcomings:
- The gravitational force of the squeegee does not act in the pin, but in the center of gravity of the squeegee,
- The force from the fluid pressure acts not only on the squeegee but also on the pump suction,
- The model completely ignores the influence of the frictional forces of the sealing surfaces in the axial direction.
The terminology used in the paper does not correspond at all to the standards defined for pumping technology:
Powered pump?
Pressure vibration?
Import and export (suction / discharge)?
Pressure energy.
Formally, the text contains many inaccuracies (Fig. 2, exchanged suction / discharge, etc.)
Author Response

(The authors gave the same response as above.)

Round 2
Reviewer 3 Report
The authors have adequately responded to all my questions and made the necessary changes to the manuscript. I am satisfied with the improvement. However, the reference that was added was not quoted properly, please check it and correct it.
Author Response
Thank you again for your comments and suggestions. I have corrected the problems you raised in the reference section of the paper.
Reviewer 4 Report
I have no further comments.
Author Response
Thank you again for your comments and suggestions. Best wishes to you.